# County-Wide Mortality Assessments Attributable to PM_2.5_ Emissions from Coal Consumption in Taiwan

**DOI:** 10.3390/ijerph19031599

**Published:** 2022-01-30

**Authors:** Chia-Pin Chio, Wei-Cheng Lo, Ben-Jei Tsuang, Chieh-Chun Hu, Kai-Chen Ku, Yi-Sheng Wang, Yung-Jen Chen, Hsien-Ho Lin, Chang-Chuan Chan

**Affiliations:** 1Innovation and Policy Center for Population Health and Sustainable Environment, College of Public Health, National Taiwan University, Taipei 10055, Taiwan; hsienho@ntu.edu.tw; 2Institute of Environmental and Occupational Health Science, College of Public Health, National Taiwan University, Taipei 10055, Taiwan; 3Graduate Institute of Epidemiology and Preventive Medicine, College of Public Health, National Taiwan University, Taipei 10055, Taiwan; nicholaslo@tmu.edu.tw (W.-C.L.); chiehchunhu@gmail.com (C.-C.H.); 4Institute of Statistical Science, Academia Sinica, Taipei 11529, Taiwan; 5Master Program in Applied Epidemiology, College of Public Health, Taipei Medical University, Taipei 11031, Taiwan; 6Department of Environmental Engineering, Innovation and Development Center of Sustainable Agriculture (IDCSA), National Chung-Hsing University, Taichung 40227, Taiwan; tsuang@nchu.edu.tw (B.-J.T.); dodo1654@email.nchu.edu.tw (K.-C.K.); y.wang2@student.tue.nl (Y.-S.W.); 7Greenpeace East Asia, Taipei 10045, Taiwan; ychen@greenpeace.org

**Keywords:** fine particulate matter (PM_2.5_), global burden of disease (GBD), coal-fired power plant (CP), combined heat and power plant (CHP)

## Abstract

Over one-third of energy is generated from coal consumption in Taiwan. In order to estimate the health impact assessment attributable to PM_2.5_ concentrations emitted from coal consumption in Taiwan. We applied a Gaussian trajectory transfer-coefficient model to obtain county-wide PM_2.5_ exposures from coal consumption, which includes coal-fired power plants and combined heat and power plants. Next, we calculated the mortality burden attributable to PM_2.5_ emitted by coal consumption using the comparative risk assessment framework developed by the Global Burden of Disease study. Based on county-level data, the average PM_2.5_ emissions from coal-fired plants in Taiwan was estimated at 2.03 ± 1.29 (range: 0.32–5.64) μg/m^3^. With PM_2.5_ increments greater than 0.1 μg/m^3^, there were as many as 16 counties and 66 air quality monitoring stations affected by coal-fired plants and 6 counties and 18 monitoring stations affected by combined heat and power plants. The maximum distances affected by coal-fired and combined heat and power plants were 272 km and 157 km, respectively. Our findings show that more counties were affected by coal-fired plants than by combined heat and power plants with significant increments of PM_2.5_ emissions. We estimated that 359.6 (95% CI: 334.8–384.9) annual adult deaths and 124.4 (95% CI: 116.4–132.3) annual premature deaths were attributable to PM_2.5_ emitted by coal-fired plants in Taiwan. Even in six counties without power plants, there were 75.8 (95% CI: 60.1–91.5) deaths and 25.8 (95%CI: 20.7–30.9) premature deaths annually attributable to PM_2.5_ emitted from neighboring coal-fired plants. This study presents a precise and effective integrated approach for assessing air pollution and the health impacts of coal-fired and combined heat and power plants.

## 1. Introduction

In Taiwan, coal is the main source of electricity generated by Taipower Company, independent power plants, combined heat and power (CHP) plants, and industrial plants (such as steel, cement, chemical, and paper plants) [1]. In 2000, total coal consumption was 45.5 MT (mega-tons) per year, with about 32.2 MT per year used in coal-fired (CP) and CHP plants. Coal-fired plants alone accounted for 70.7% of total coal consumption. Over a recent five-year period (2013–2017), total average coal consumption increased to 66.7 MT per year (about 1.5-fold increase), with coal consumption for CHP plants reaching 44.7 MT per year (about 1.4-fold increase). At about 67.0%, the two types of power plants together accounted for a steady proportion of total coal consumption over the five years. A statistical report from the Bureau of Energy of the Ministry of Economic Affairs of the Republic of China (Taiwan) showed that 100% of the coal consumed in the country has been imported from other countries since 2001 [2].

The coal-fired process may emit various pollutants, including particulate matter (PM), sulfur oxides (SOx), nitrogen oxides (NOx), carbon monoxide (CO), carbon dioxide (CO_2_), and volatile organic compounds, and even persistent organic pollutants (POPs, including PAHs, dioxins, and furans) [3,4,5]. Based on Taiwan’s Emissions Database System (TEDS) version 9, which was built in 2013, five coal-fired plants with 21 generator sets emitted 2482.15, 32,081.25, and 46,475.88 T/y (tons per year) of PM_2.5_ (PM with aerodynamic diameter less than 2.5 μm), SO_2_, and NOx, respectively. In addition, 33 combined heat and power plants with 65 generator sets emitted a total of 877.39, 9732.7, and 23,839.7 T/y of PM_2.5_, SO_2_, and NOx, respectively [6].

Previous studies showed that PM_2.5_ causes morbidity and mortality from cardiovascular, respiratory, and central nervous system diseases and cancers [4,7,8,9,10,11,12,13,14,15,16]. Some studies also looked at emergency visits and hospital admissions for PM-related diseases [17,18,19,20].

To estimate the mortality attributable to PM_2.5_ exposures, Burnett et al. [12] integrated several disease-specific exposure-response functions (or mortality relative risk functions), including ischemic heart disease (IHD), cerebrovascular disease (stroke), lung cancer (LC), and chronic obstructive pulmonary disease (COPD), based on the Global Burden of Disease (GBD) study design. The GBD originally used a comparative risk assessment (CRA) framework to estimate the mortality burden attributable to each risk factor at the country level [14,21]. Lo et al. [22] applied this approach to estimate the national and sub-national numbers of deaths attributable to long-term monitored PM_2.5_ exposures in Taiwan. Hwang and co-authors showed the air quality region-based PM_2.5_ exposures and spatiotemporal assessments of mortality attributable to ambient PM_2.5_ by using AirQ+ software version 1.0 released by the World Health Organization (WHO) during 2008–2015 in Taiwan [23,24]. Koplitz et al. [25] combined modeled air quality PM_2.5_ and ozone exposures with the GBD approach to estimate adult deaths and premature deaths attributable to rising coal-fired power plant emissions in Southeast Asia. Recently, Chio et al. [26] successfully assessed the long-term cumulative deaths attributable to PM_2.5_ emitted from a single planned coal-fired power plant in Taiwan, based on an air quality model and GBD approach.

With Taiwan’s increasing use of coal for electricity generation, pollutants, such as PM and precursor gases, are expected to rise. Because of this trend toward the use of coal, it is important to evaluate air quality and its associated impact on mortality. To achieve the goals of this study, we applied a local and high spatial-resolution air quality model to estimate PM_2.5_ increments and employed the GBD approach to estimate the number of adult deaths and premature deaths attributable to the modeled PM_2.5_ exposures.

## 2. Materials and Methods

### 2.1. County-Level PM_2.5_ Estimates Using GTx Modeling

We applied the Gaussian trajectory transfer-coefficient (GTx) model to estimate PM_2.5_ concentrations emitted from coal-fired and combined heat and power plants. The GTx model is composed of an air dispersion model based on a Lagrangian trajectory framework along with Gaussian dispersion theory [27,28,29,30,31]. In this study, the GTx model was executed in backward-trajectory modes (traj.79.LatLon and pm.1.010). Meanwhile, we collected the emissions inventories of the 5 coal-fired and 33 combined heat and power plants from the TEDS 9.0 database, as presented in Table 1 [6]. Figure 1 shows the locations of the 5 CP and 33 CHP plants in Taiwan. A total of 21 sectors were installed in CP plants and 65 combined heat and power plants. As we know, one plant (CP#01) used an ultra-supercritical boiler, three plants (i.e., CP#02, CP#03, and CP#04) used a supercritical boiler, and one plant (CP#05) used a subcritical boiler (shown in Appendix A) [32,33]. One coal-fired plant and nine combined heat and power plants were located in Kaohsiung City. Table 1 shows that the most coal-fired plant emissions came from a plant in Taichung City, which accounted for 1235.44 T/y of Emi_PM_2.5_ emissions, 14,886.12 T/y of Emi_SO_2_, 23,188.26 T/y of Emi_NOx, and 63,197.19 T/y of Emi_PM_2.5__eq emissions (i.e., equivalent PM_2.5_ emissions), respectively. Overall, there were 2482.15 T/y of PM_2.5_, 32,081.25 T/y of SO_2_, 46,475.88 T/y of NOx, and 130,607.30 T/y of PM_2.5__eq emitted from CP plants in five cities or counties. In Yunlin County, there were 306.23 T/y of Emi PM_2.5_, 3418.55 T/y of Emi_SO_2_, 6458.40 T/y of Emi_NOx, and 16,288.36 T/y of Emi_PM_2.5__eq emissions from 4 CHP plants. CHP plants in 12 cities/counties emitted a total of 877.39 T/y of PM_2.5_, 9732.71 T/y of SO_2_, 23.839.70 T/y of NOx, and 54,812.10 T/y of PM_2.5__eq. That is to say, there was no power sector for CP and CHP plants located in Keelung City, Taipei City, Hsinchu City, Nantou County, Chiayi City, and Pingtung County (Table 1). Meanwhile, the formula for the equivalent PM_2.5_ emissions was as follows:(1)$$\mathrm{Emi}\_{\mathrm{PM}}_{2.5}\_\mathrm{eq}\;(\mathrm{using}\;\mathrm{molecular}\;\mathrm{weight})=\;\;0.85\;\times \;\mathrm{Emi}\_{\mathrm{SO}}_{2}\;\times \;\frac{\mathrm{MW}\_{\left({\mathrm{NH}}_{4}\right)}_{2}{\mathrm{SO}}_{4}}{\mathrm{MW}\_{\mathrm{SO}}_{2}}+0.58\;\times \;\mathrm{Emi}\_{\mathrm{NO}}_{x}\;\times \;\frac{\mathrm{MW}\_{\mathrm{NH}}_{4}{\mathrm{NO}}_{3}}{\mathrm{MW}\_\mathrm{NO}}+\mathrm{Emi}\_{\mathrm{PM}}_{2.5}$$
where all the SO_2_ and NOx are transformed to ammonium sulfate ((NH_4_)_2_SO_4_) and ammonium nitrate (NH_4_NO_3_) with enough ambient ammonium ions (NH_4_^+^). The fine fractions of PM_10_ are 0.85 for (NH_4_)_2_SO_4_ and 0.58 for NH_4_NO_3_ compounds [28]. These two ratios were based on the field measured data in a coastal area of Taiwan from Tsai and Cheng [34].

All parameters used in the GTx model are listed in Table 2. Using settings for the year 2013, we also collected observations from 117 meteorological datasets and 71 air quality monitoring networks (AQMNs). We modeled the PM_2.5_ at 71 AQMNs for all coal-using plants. In the “traj” mode of the GTx model, wind field data were obtained by applying a weighted interpolation method to meteorological data, and the pollutant concentrations were modeled in the “pm” mode of the GTx. Here, we estimated a modeling period of 168 h over one whole year, 2013, using the GTx model. The model’s assumptions are shown in Table 1, including emissions data treatments, background concentrations from overseas, and dry deposition velocity of NOy.

We used three indices in the validation criteria for the air quality model [35] to check our PM_2.5_ estimates against PM_2.5_ observations from the TEPA AQMNs for the year 2013 as follows.

Mean fractional bias (MFB) for PM_2.5_ model validation should fall in intervals of ±35%.


(2)
$$\mathrm{MFB}=\frac{2}{M\times N}\sum_{k=1}^{M}\sum_{i=1}^{N}\left(\frac{P_{i,k}-O_{i,k}}{P_{i,k}+O_{i,k}}\right)$$


2.The mean fractional error (MFE) for PM_2.5_ model validation should be in intervals less than 55%.


(3)
$$\mathrm{MFE}=\frac{2}{M\times N}\sum_{k=1}^{M}\sum_{i=1}^{N}\left|\frac{P_{i,k}-O_{i,k}}{P_{i,k}+O_{i,k}}\right|$$


3.The correlation coefficient (R) for PM_2.5_ model validation should be higher than 0.5.


(4)
$$R=\frac{1}{M\times N}\sum_{k=1}^{M}\sum_{i=1}^{N}\left[\frac{\left(P_{i,k}-\bar{P}\right)\left(O_{i,k}-\bar{O}\right)}{S_{P}S_{O}}\right]$$


### 2.2. Estimation of the Population-Attributable Fraction of Mortality Attributable to PM_2.5_

Our study considered the mortality from four major diseases defined according to the International Classification of Diseases, Tenth Revision (ICD-10) codes. These include ischemic heart diseases (denoted as IHD, I20–I25), stroke (I60–I67, I69.0, I69.1, I69.2, and I69.3), lung cancer (denoted as LC, C33, and C34), and chronic obstructive pulmonary disease (denoted as COPD, J40–J44). We estimated the population-attributable fraction (PAF) of cause-specific mortality associated with PM_2.5_ for each disease by year and by county level.

According to previous longitudinal studies and our GBD 2013 nonlinear exposure-response model, we assumed a linear relationship between PM_2.5_ exposure (moderate exposure level: <50 μg/m^3^ annual average) and the corresponding disease outcomes to derive relative risk (RR) estimates for our analyses [12]. The risk effect sizes were estimated by 10 μg/m^3^ increments of PM_2.5_ for each disease outcome. These RR estimates for 4 disease outcomes were calculated by 10 μg/m^3^ PM_2.5_ increment. The standard level of PM_2.5_ in our draft is recommended by the World Health Organization (10 μg/m^3^) [22]. For the uncertainty interval of our RR estimation, we adopted data by meta-analysis of the results of previous longitudinal studies, assuming a constant association between point estimates and standard errors in our RR estimation processes to allow for a monotonic relation between PM_2.5_ and health outcomes and to obtain a conservative confidence interval (CI) [36,37,38,39,40,41].

The PAF measures what proportion of the disease burden in a given population would be prevented or postponed if the PM_2.5_ exposure level were shifted to an alternative optimal exposure level. We used the following formula to calculate the PAF of cause-specific mortality for a specific disease at the county level:(5)$$PAF_{i,j}=\frac{RR_{c\left(i\right),j}-1}{RR_{c\left(i\right),j}}$$
where *c(i)* is the estimated 10-year average level of PM_2.5_ in county *i*, which reflects the cumulative exposure of PM_2.5_, and *RR**c(i*), *j* is the RR for disease *j* at exposure level *c(i)*, as determined using our linear model. We further multiplied the PAF by the cause-specific number of deaths to obtain the mortality burden attributable to PM_2.5_, as seen in the following equation [25]:(6)$$\Delta M_{\alpha ,k}=y_{0_{\alpha ,k}}\left(1-e^{-\beta_{\alpha }\Delta x_{k}}\right)P_{k}$$
where Δ*M*_α,k_: the change in annual mortality (or pre-mature mortality) [deaths per year] due to coal pollution for each cause of death *α* in each city/county *k*; *y*_0_*_α,k_*: is the cause-specific baseline death rate or PAF [% per year] in the city/county; Δ*x* is the population-weighted change in PM_2.5_ concentration [μg/m^3^]; *β* is the cause-specific concentration-response function (CRF) relating a one-unit change; *P_k_* is the total population of the city/county *k*. All the analyses included age, gender, and county. Meanwhile, the age group for adult deaths included the population over 25 years old, and the age group for premature deaths included the population between 25 and 70 years old.

### 2.3. Uncertainty and Sensitivity Analyses

Statistical simulation was performed to deal with the uncertainty induced by sampling variability. In the model, we randomly sampled 1000 sets of PM_2.5_ exposures and corresponding RRs from prior normal distributions of PM_2.5_ concentrations and RRs. Each set of sampled PM_2.5_ concentrations and RRs was used to estimate the PAF and the amount of PM_2.5_ for adult deaths (or premature deaths) for each county separately, by age group. We ranked the resulting 1000 PAFs and the number of PM_2.5_-attributable deaths and considered the corresponding 2.5th and 97.5th percentiles as the range for 95% CIs.

## 3. Results

### 3.1. Comparison of PM_2.5_ Observation and Estimates by GTx Modeling

The PM_2.5_ concentrations estimated by the GTx model were cross-validated with in situ observations in our previous study [26]. In brief, overall, 25,684 daily-site data pairs showed the MFB, MFE, and R indices in Taiwan were 7%, 37%, and 0.53, respectively, and they satisfied the model criteria [35]. For spatial comparison of PM_2.5_, the modeled PM_2.5_ findings were higher than observed PM_2.5_ findings in mountain areas in northern, central, and eastern Taiwan and the coastal area in Miaoli County. Conversely, the model PM_2.5_ figures were lower than the observed PM_2.5_ figures in southern Taiwan (Figure 2A,B).

### 3.2. Spatial Distribution of Population and PM_2.5_ Observations in Taiwan

At the end of 2013, there were a total of 23,373,517 people residing in Taiwan. Figure 3A shows the citywide distribution of the population in Taiwan. New Taipei City is a tier 1 level (of 5 grades), with 3,954,929 people. Kaohsiung City, Taichung City, Taipei City, and Taoyuan City were listed in the tier 2 level, with residents numbering 2,779,877, 2,701,661, 2,686,516, and 2,044,023, respectively [42]. Moreover, the spatial distribution in cities and counties based on 10-year averaged PM_2.5_ concentrations is shown in Figure 3B. Meanwhile, the overall average of observed PM_2.5_ concentration with standard deviation (SD) was 30.59 ± 4.05 μg/m^3^. For spatial variation, in Chiayi and Kaohsiung cities, both listed in the tier 1 level (of 5 grades), the PM_2.5_ averages were higher than 44.0 μg/m^3^. In addition, Tainan City, Changhua County, Nantou County, Yunlin County, and Chiayi County are listed in the tier 2 level (ranging between 35 and 40 μg/m^3^), and Hsinchu City, Taichung City, and Pingtung County are listed in the tier 3 level (ranging between 30 and 35 μg/m^3^). For GTx-modeled PM_2.5_ estimates from all coal-fired power plants (namely, CP and CHP plants), Nantou County is still listed in the tier 1 level (range between 4.0 and 6.0 μg/m^3^) for PM_2.5_ concentration modeling. However, Miaoli County, Changhua County, Yunlin County, Chiayi County, Chiayi City, Tainan City, and Kaohsiung City are listed in the tier 2 level (range between 2.0 and 4.0 μg/m^3^) (Figure 3C and Table 3).

In order to evaluate the city/county based PM_2.5_ estimates in different coal use model scenarios (Table 3), results ranged from 13.66 μg/m^3^ to 44.39 μg/m^3^ with an average of 30.59 μg/m^3^ of city/county based on 10-year average PM_2.5_ exposures. Results from all coal-fired plants also show that it ranged from 0.32 μg/m^3^ to 5.46 μg/m^3^ increments with an estimated PM_2.5_ average of 2.03 μg/m^3^. In Table 3, the results show that Nantou and Miaoli Counties had the highest PM_2.5_ concentrations from total CP and CHP plants, respectively, by using GTx modelings.

### 3.3. Counties Affected by PM_2.5_ from CP and CHP Plants and Maximum Distance

The “maximum (affected) distance” was calculated by using the Euclidian distance method between each CP or CHP plant and AQMN under different modeled PM_2.5_ levels (shown in Appendix A). In our models, results ranged from 4 to 16 counties and from 12 to 66 AQMNs affected by coal-fired plants, with 0 to 6 counties and 0 to 18 AQMNs affected by combined heat and power plants under increments greater than 0.1 μg/m^3^ PM_2.5_. The maximum affected distances ranged from 155 to 272 km for coal-fired plants and from 0 to 157 km for combined heat and power plants (Table 4). The CP#02 plant, located in Taichung City, had the maximum air quality impact affecting 16 counties and 66 AQMNs. In this case, the distance from CP#02 plant to Linyuan AQMN in the city of Kaohsiung was 193 km. In addition, CP#04 had affected the maximum distance of 272 km, traveling from the CP#04 plant in Kaohsiung City to Xindian AQMN in New Taipei City (Table 4). Under the same conditions, however, the CHP#27 plant in Yuanlin County also had the maximum air quality impact, affecting 6 counties and 18 AQMNs. In that case, the distance from the CHP#27 plant to Longtan AQMN in Taoyuan City was 157 km. In addition, there were 14 combined heat and power plants that did not affect any AQMN due to PM_2.5_ increments less than 0.1 μg/m^3^ (Table 4). The PM_2.5_ impact from coal-fired and combined heat and power plants differed significantly due to emissions rates, and the stack and flow parameters. Using the same approach, the maximum affected distances were 306 km from CP#01 in New Taipei City and 163 km from CP#02 in Taichung City, at 0.01, and 1.0 μg/m^3^ PM_2.5_ increments, respectively. Additionally, the maximum affected distances were estimated to be 284 km from CHP#07 in Taoyuan City and 3 km from CHP#23 in Miaoli County at 0.01 and 1.0 μg/m^3^ PM_2.5_ increments, respectively (Results not shown).

### 3.4. Assessment of Adult Deaths and Premature Deaths Attributable to PM_2.5_

The overall PAF estimate for total deaths was 19.6% (range: 9.6–25.8%) and for premature deaths was 25.5% (range: 12.9–32.1%). Nationally, the PAF for PM_2.5_ causing the four diseases was 16.9% for IHD (range: 9.3–22.7%), 26.7% for stroke (range: 13.2–35.6%), 16.7% for LC (range: 7.5–22.3%), and 13.7% for COPD (range: 6.2–18.6%). However, the figures for premature deaths were 24.2% for IHD (range: 12.3–31.9%), 39.0% for stroke (range: 20.1–49.2%), 16.8% for LC (range: 7.5–22.3%), and 14.1% for COPD (range: 6.4–18.3%) (Appendix A). The overall adult deaths estimated per year were 1483.8 for IHD (95%CI: 1413.7–1558.8), 2996.8 for stroke (95%CI: 2886.8–3108.5), 1455.68 for LC (95%CI: 1404.3–1504.2), 736.7 for COPD (95%CI: 564.6–908.7), and 6672.9 for total deaths (95%CI: 6441.9–6921.6) (Appendix A).

The estimated adult deaths per year due to PM_2.5_ from all coal-fired plants (Figure 4A) ranked New Taipei, Taichung, Tainan, and Kaohsiung Cities as tier 1 level (of 7 grades), with a range of 30 to 60 deaths per year. Taoyuan City, Changhua County, Nantou County, Yunlin County, and Chiayi County ranked in the tier 2 level with a range of 20 to 30 deaths per year, and Taipei City, Ilan County, and Pingtung County ranked in the tier 3 level with a range from 15 to 20 deaths per year. More details for disease-specific adult deaths attributable to PM_2.5_ from all coal-fired plants are shown in Appendix A. The estimated adult deaths per year attributable to PM_2.5_ from all coal-fired plants were 81.7 for IHD (95%CI: 75.3–87.9), 154.4 for stroke (95%CI: 143.2–166.0), 81.4 for LC (95%CI: 76.4–86.7), 42.0 for COPD (95%CI: 32.5–52.0), and 359.6 for total deaths (95%CI: 334.8–384.9).

The overall estimated premature deaths per year were 602.6 for IHD (95%CI: 571.3–632.6), 1.174.8 for stroke (95%CI: 1.134.0–1214.6), 609.0 for LC (95%CI: 584.3–633.3), 75.4 (95%CI: 59.4–91.8) for COPD, and 2461.7 total deaths (95%CI: 2401.1–2529.1) (Appendix A). Estimated premature deaths per year attributable to all coal-fired plants (Figure 4B) showed New Taipei City ranked in tier 1 (of 7 grades) with a range of 15 to 25 deaths per year. New Taipei, Taichung, and Tainan Cities ranked as tier 2 with a range from 10 to 15 deaths per year, and Taoyuan City, Yunlin County, and Chiayi County ranked in the tier 3 level with a range from 7.5 to 10 deaths per year. More details for disease-specific premature deaths attributable to PM_2.5_ from all coal-fired plants are shown in Appendix A. Estimated premature deaths per year attributable to PM_2.5_ from all coal-fired plants were 31.2 for IHD (95%CI: 29.0–33.5), 54.9 for stroke (95%CI: 51.3–58.7), 34.1 for LC (95%CI: 31.9–36.4), 4.2 for COPD (95%CI: 3.4–5.0), and 124.4 total deaths (95%CI: 116.4–132.3).

## 4. Discussion and Limitations

### 4.1. GTx Modeling

The air quality modeling results depend completely on emissions and meteorological data. In our study, the GTx model for PM_2.5_ concentrations used the emissions inventories from CP and CHP plants (from TEDS 9.0) and observed meteorological data (from Taiwan’s Environmental Protection Agency and the Central Weather Bureau in 2013) (Table 2). Before scenario modeling through the GTx model, we first verified the modeled results using PM_2.5_ observations obtained from 71 Taiwan EPA AQMNs. If we double checked the comparisons with 71 individual AQMS sites, the results showed that 83% (59/71) satisfied the comparison criteria for MFB, 93% (66/71) for MFE, and 89% (63/71) for R (data not shown). In our models, there were still many over- and underestimates related to geography in Taiwan. The overestimated regional statistics were on average 5.3, 1.4, and 7.3 μg/m^3^ higher in northern, central, and eastern Taiwan, respectively. Underestimated figures were 3.3 and 2.3 μg/m^3^ lower in the Yun-Chia area and southern Taiwan [26]. Several possible reasons might result in the differences between models and observations, including (1) the possibility that emissions data did not capture the correct condensable fraction of PM emitted from stationary sources [43,44]; (2) secondary aerosol growth mechanisms might be more complex than model defaults [28,29,30,31]; (3) the emissions data for all sources might not be revised simultaneously, as shown in Table 2; (4) hourly emissions from sources were unknown; and (5) the background concentrations of PM might be misused.

According to a previous report [45], 37% of PM_2.5_ came from outside of Taiwan, and 18.2%, 21.2%, and 23.5% of PM_2.5_ were contributed from point, line, and area sources, respectively. The report also indicated that about 4.5% of PM_2.5_ contributed from power plants and 13.7% of PM_2.5_ contributed to other industrial sources throughout Taiwan. However, there is no detailed county-based PM_2.5_ source apportionment study with source/trajectory or receptor models in Taiwan. In our analysis with the GTx model, we estimated the median county-averaged contribution of 4.1% (range: 1.4–12.0%) of PM_2.5_ from 5 coal-fired power plants. If taking into account 5 coal-fired power plants (CPs) and 33 coal-fired combined heat and power plants (CHPs), the median county-average contribution was 6.1% (range: 2.4–16.1%) of PM_2.5_ (shown in Appendix A). Our PM_2.5_ estimates from 5 CPs and/or 5 CPs + 33 CHPs indicated good agreement with the previous report.

In particular, results showed that there were no CP and CHP plants located in Keelung City, Taipei City, Hsinchu City, Nantou County, Chiayi City, and Pingtung County (Table 1), yet CP or CHP plants contributed 1.09, 0.87, 1.40, 5.64, 2.79, and 1.87 μg/m^3^ to annual average PM_2.5_ (Table 3). The highest county-level estimate (5.64 μg/m^3^) was in Nantou County, with emissions from CP and CHP plants accounting for almost 16% of the 10-year averaged PM_2.5_ concentration. Results from GTx modeling in Nantou County also showed that nearly 75% (4.22 of 5.64 μg/m^3^) of PM_2.5_ concentrations were emitted from coal-fired plants(Table 3). Nantou County, the only inland district in Taiwan, accounted for 95% of the total area over 100 m high (including 66% of the total area above 1000 m) [46]. High altitude terrain, geographic basin structures, and monsoons might be leading factors causing low dispersion and accumulation effects when the various pollutants travel via air parcels into Nantou County from neighboring cities or counties to the west (Figure 1) [47].

### 4.2. Assessing Health Impacts

We applied the GBD approach to estimate the adult deaths and premature deaths attributable to PM using different age intervals in our study because the disease-specific mortalities were age dependent. The results indicated that there were significant differences in PAFs for adult deaths and premature deaths for IHD (16.9% of adult deaths vs 24.2% of premature deaths) and stroke (26.7% of adult deaths vs 39% of premature deaths) (Appendix A). Unlike cardiovascular system diseases (IHD and stroke), the differences between PAFs for adult deaths and premature deaths due to the effect of PM on the respiratory system (LC and COPD) showed no gaps. In 2013, strokes were the leading cause of adult deaths and premature deaths attributable to ambient PM_2.5_ exposure in Taiwan, accounting for an estimated 44.9% (2996.8 of 6672.9 per year) of adult deaths and 47.7% (1174.8 of 2461.7 per year) of premature deaths on average. Compared with a previous study [22], our estimate of total deaths fell reasonably into the interval (range: 6282 to 7869 deaths per year on average) between the main analysis and a nine-year time lag. We extracted the estimates in 2013 from the study results of Hwang et al. [23], and the deaths attributable to PM_2.5_ ranged from 2026 to 7066 deaths (median: 4665 deaths) per year. Our assessments also fell into estimated intervals but with more accuracy. In terms of geographic analysis, Kaohsiung City was the city with the largest impact on total deaths assessed, with PM emissions estimated to account for 15.7% of adult deaths (1046.3 of 6672.9 per year) and 16.7% of premature deaths (410.3 of 2461.7 per year) on average (Appendix A).

Total adult and premature death estimates attributable to PM in Taiwan were 359.6 (95%CI: 334.8–384.9) and 124.4 (95%CI: 116.4–132.3) deaths per year, respectively. Actually, Nantou County was the district with the most severe health impact for a non-power sector installed county; total estimated adult deaths and premature deaths per year were 28.7 (95%CI: 14.5–43.9) and 9.7 (95%CI: 5.1–14.4), respectively (Appendix A). Koplitz et al. [25] reported 109 premature deaths per year, which was slightly lower than our estimated interval (124.4 (95%CI: 116–132.3) deaths per year). Furthermore, our health assessments showed 75.8 (95%CI: 60.1–91.5) adult deaths and 25.8 (95%CI: 20.7–30.9) premature deaths attributable to PM_2.5_ emitted from coal-fired plants in counties without power plants, which included Keelung City, Taipei City, Hsinchu City, Nantou County, Chiayi City, and Pingtung County (data not shown).

### 4.3. Limitations

There were several limitations to our study. First, to reasonably link PM_2.5_ exposures to health outcomes, we used 10-year PM_2.5_ averages for a long-term exposure level capable of promoting or aggravating disease. Second, following previous studies, we only included IHD, stroke, lung cancer, and COPD [12,14,22,25]. Actually, many other potential health impacts were not included in our estimates, such as hospital admissions, emergency department visits, doctor visits, restricted activity/reduced performance, medication use, symptoms, physiological changes in the cardiovascular system, impaired pulmonary function, and subclinical effects proposed by previous reports [48]. Lastly, our GTx modeling might contain several uncertainties in its current version, including incomplete or missing emissions data, background concentrations, and secondary aerosol growth mechanisms, as mentioned above. These uncertainties in each AQMN might cause positive or negative effects on PM_2.5_ modeling. Furthermore, we checked an overall 25,684 daily-site data pairs with observed and GTx modeled PM_2.5_ from Taiwan, and the results showed that the MFB, MFE, and R indices satisfied the model criteria [35]. Therefore, we believe that our estimates are credible. Indeed, most air quality modeling studies selected the Weather Research and Forecasting Model coupled with Chemistry (WRF-Chem) based model system because of its excellent meteorological parameter forecasting. However, our model system (GTx model) actually used real meteorological data from 117 sites, as shown in Table 2. Our GTx model was mainly developed for Taiwan. Although the GTx model selected more simple photochemical reactions for secondary PM_2.5_ productions from gaseous pollutants, the final estimates for PM_2.5_ were acceptable for us with several validations.

In fact, the population spatial distribution represented the urbanization levels, and all coal-fired power plants emitted PM_2.5_ spatial distributions just represented the air quality levels by those major point sources (Figure 3). These two levels did not match originally. Tsai et al. [49] and Lu et al. [50] used different receptor models (Positive Matrix Factorization and Chemical Mass Balance, respectively) to analyze different coastal areas (Taichung and Tainan, respectively). Therefore, different major sources (road dust in Taichung and traffic emissions in Tainan, respectively) might be possible. Actually, the above two references for PM_2.5_ source apportionments were totally different from our study. Our study framework was based on the air quality model (GTx model) and the global burden of disease approach. In fact, any air quality could replace the GTx model. As I know, if the entire emission data, meteorological data, and geological data were fully prepared for GTx modeling in other countries. Our method could be applied outside of Taiwan. Indeed, a previous report showed that the line or mobile source contribute 21.2% of PM_2.5_ in Taiwan [45]. Therefore, transportation systems should be discussed, including electronic transport tools [51,52,53].

In our analyses, we did not take into account the ambient concentrations and health impacts of CO, POPs, and others. Yang et al. [54] provided the short-term adverse effects of air pollution and concluded that CO increases hospital admissions for cardiovascular diseases. Raub et al. [5] reviewed the sources of CO indoors/outdoors, such as in workplace/home with poor ventilation and street intersections/internal combustion engines/industrial sources, respectively. The complications of CO poisoning were also reviewed, including immediate death, myocardial impairment, hypotension, arrhythmias, and pulmonary edema. Previous studies showed that several congeners of dioxins/furans could be found in gaseous pollutants emitted from CP or CHP plants [55,56].

## 5. Conclusions

Based on our GTx modeling, PM_2.5_ contributed from coal-fired and combined heat and power plants ranged between 0.32 and 5.64 μg/m^3^, with an average of 2.03 ± 1.29 μg/m^3^ in city/county-based levels. The PM_2.5_ contribution from all coal-fired plants was greatest in Nantou County, estimated at 5.64 ± 1.37 μg/m^3^, equal to 15% of the 10-year PM_2.5_ average level. Our results showed coal-fired plants always contributed more PM_2.5_ than combined heat and power plants in affected counties. Of the overall average PM_2.5_ in Taiwan, 1.33 ± 0.89 came from coal-fired plants and 0.70 ± 0.45 μg/m^3^ from combined heat and power plants. While PM_2.5_ increments greater than 0.1 μg/m^3^, coal-fired plants affected as many as 16 counties and 66 AQMNs and combined heat and power plants affected 6 counties and 18 AQMNs. The maximum distances affected by coal-fired and combined heat and power plants were 272 km and 157 km, respectively. According to the health statistics, PM_2.5_ emissions from coal-fired plants accounted for 359.6 (95%CI: 334.8–384.9) adult deaths and 124.4 (95% CI: 116.4–132.3) premature deaths per year. Our results also demonstrated severe air quality and probable health impacts in six counties without power plants. This study provides an effective and precise integrated approach to assessing air quality and associated health impacts attributable to PM_2.5_ emitted from coal-fired and combined heat and power plants.

## Figures and Tables

**Figure 1 ijerph-19-01599-f001:**
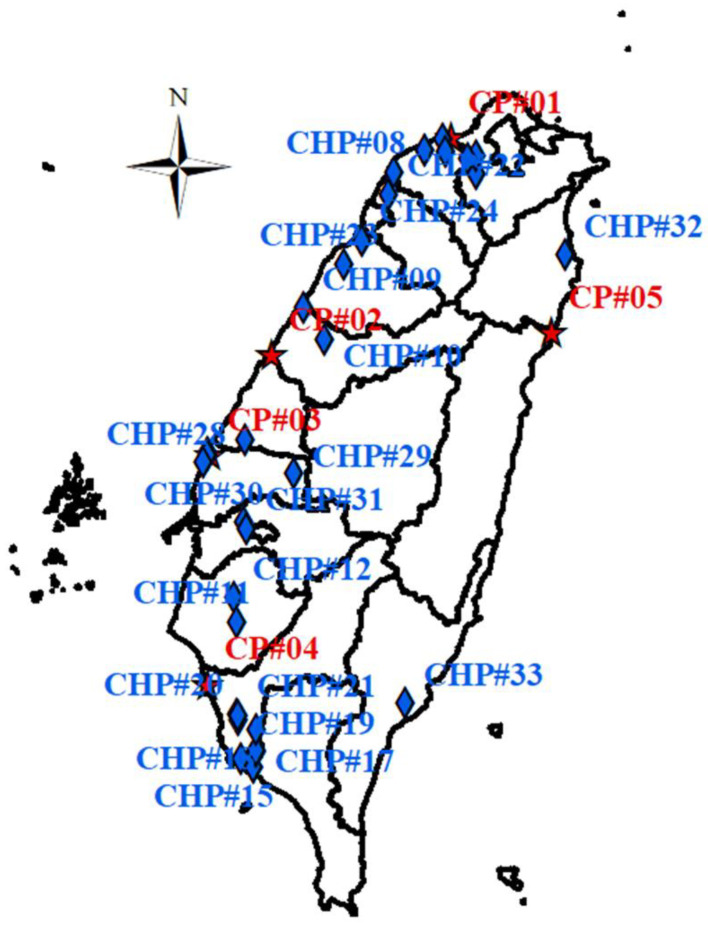
Locations of coal-fired power (CP) and combined heat and power (CHP) plants in Taiwan. A total of 21 and 65 generators were installed in 5 CP and 33 CHP plants, respectively.

**Figure 2 ijerph-19-01599-f002:**
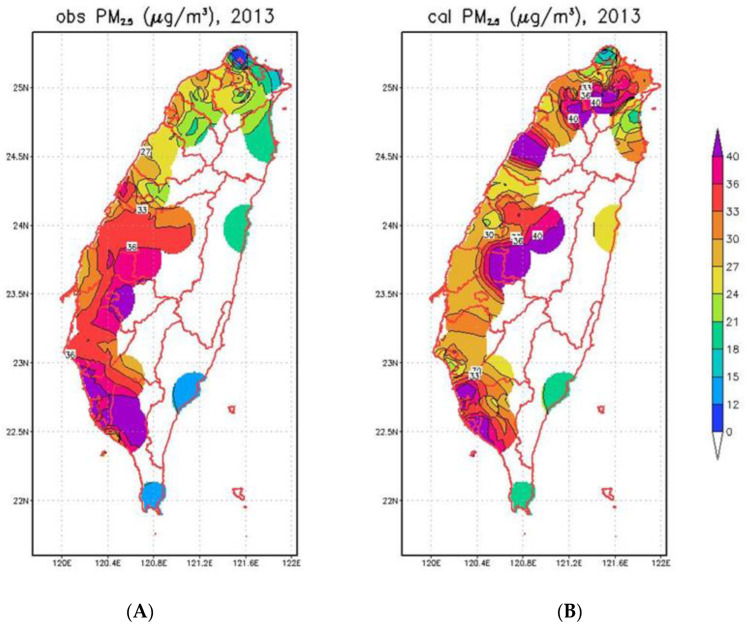
Comparison of city/county based (**A**) PM_2.5_ observation and (**B**) PM_2.5_ estimates by GTx model in 2013.

**Figure 3 ijerph-19-01599-f003:**
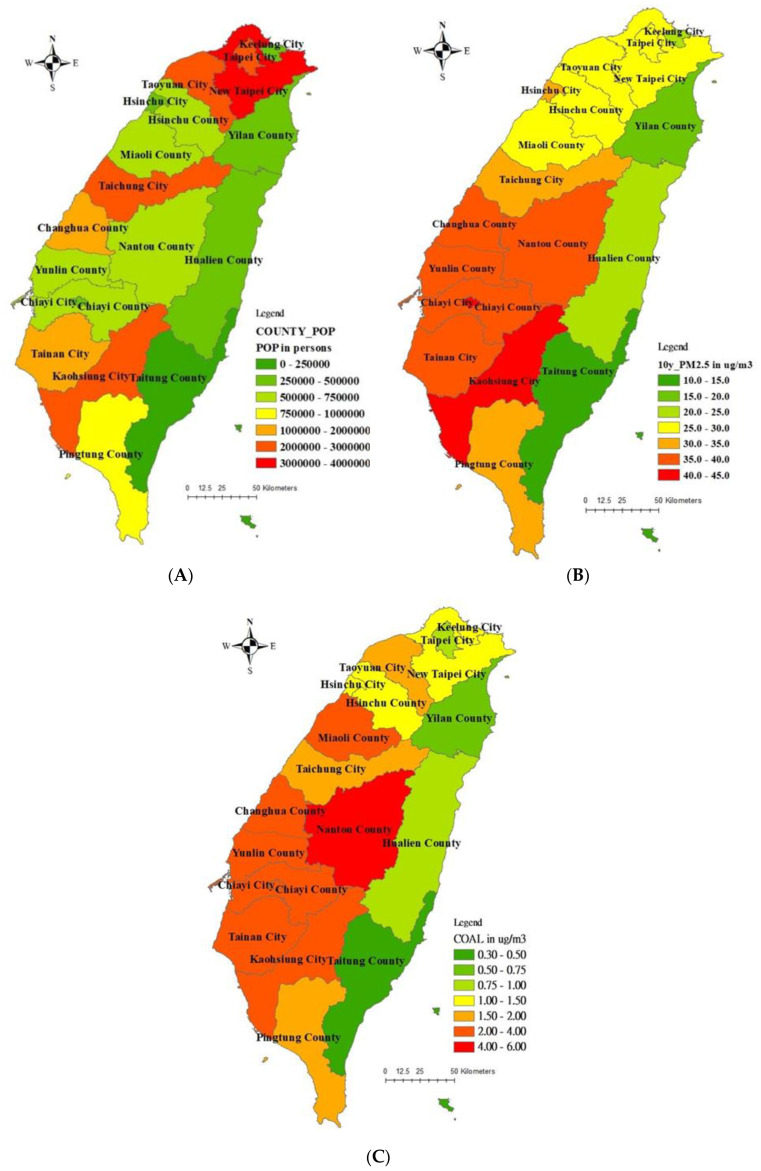
Spatial distribution of city/county based (**A**) populations, (**B**) 10-year averaged PM_2.5_ concentration, and (**C**) GTx-modeled PM_2.5_ estimates from all coal-fired power plants.

**Figure 4 ijerph-19-01599-f004:**
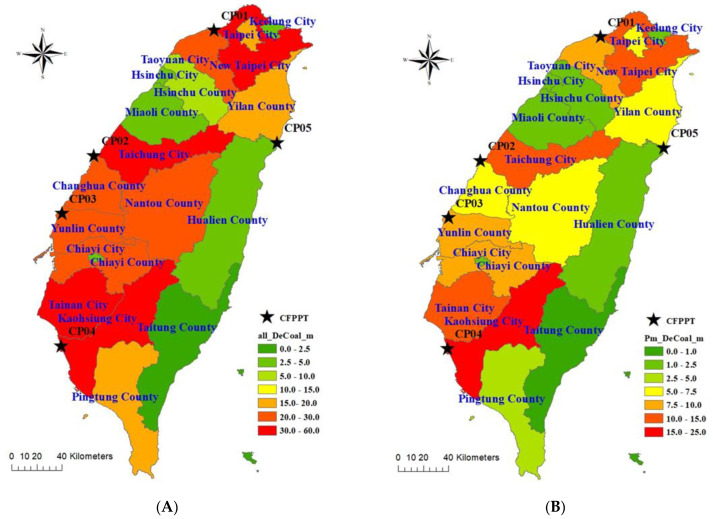
Spatial distribution of city/county based attributable (**A**) total deaths and (**B**) premature deaths due to PM_2.5_ from all coal-fired power plants.

**Table 1 ijerph-19-01599-t001:** City-/county-based emission inventories for CP and CHP plants in Taiwan.

City/County (No. of CP, No. of CHP)	Coal-Fired Power (CP) Plants [T/y]				Coal-Fired Combined Heat and Power (CHP) Plants [T/y]			
	Emi_PM_2.5_	Emi_SO_2_	Emi_NOx	Emi_PM_2.5__eq ^1^	Emi_PM_2.5_	Emi_SO_2_	Emi_NOx	Emi_PM_2.5__eq ^1^
Taipei City (0 CP, 0 CHP)	0	0	0	0	0	0	0	0
Taichung City (1 CP, 2 CHPs)	**1235.44**	**14,886.12**	**23,188.26**	**63,197.19**	98.16	402.38	1074.71	2465.81
Tainan City (0 CP, 2 CHPs)	0	0	0	0	43.96	867.92	734.83	2072.06
Kaohsiung City (1 CP, 9 CHPs)	683.09	10,836.69	13,353.18	40,334.08	158.92	2351.76	4517.76	11,269.31
Keelung City (0 CP, 0 CHP)	0	0	0	0	0	0	0	0
Hsinchu City (0 CP, 0 CHP)	0	0	0	0	0	0	0	0
Chiayi City (0 CP, 0 CHP)	0	0	0	0	0	0	0	0
New Taipei City (1 CP, 2 CHPs)	70.17	1180.68	4071.76	8437.75	4.39	171.85	1277.85	2282.07
Taoyuan City (0 CP, 6 CHPs)	0	0	0	0	53.40	1007.83	2431.44	5580.89
Hsinchu County (0 CP, 1 CHP)	0	0	0	0	10.06	110.28	518.48	1005.32
Miaoli County (0 CP, 2 CHPs)	0	0	0	0	105.24	497.23	2874.67	5423.11
Changhua County (0 CP, 1 CHP)	0	0	0	0	15.82	95.21	470.38	910.24
Nantou County (0 CP, 0 CHP)	0	0	0	0	0	0	0	0
Yunlin County (1 CP, 4 CHPs)	190.46	1930.53	3268.29	8629.86	**306.23**	**3418.55**	**6458.40**	**16,288.36**
Chiayi County (0 CP, 2 CHPs)	0	0	0	0	56.48	661.90	3155.52	6097.40
Pingtung County (0 CP, 0 CHP)	0	0	0	0	0	0	0	0
Ilan County (0 CP, 1 CHP)	0	0	0	0	17.29	91.00	249.62	562.91
Hualien County (1 CP, 0 CHP)	302.97	3247.22	2584.39	10,008.41	0	0	0	0
Taitung County (0 CP, 1 CHP)	0	0	0	0	7.45	56.80	76.02	224.60
Taiwan (5 CPs, 33 CHPs)	2482.15	32,081.25	46,475.88	130,607.30	877.39	9732.71	23,839.70	54,812.10

^1^ Detail for PM_2.5__emi equivalent was shown in the Section 2. Column maximum value is shown in bold.

**Table 2 ijerph-19-01599-t002:** Parameters used in the Gaussian trajectory transfer-coefficient (GTx) model.

Model Parameters	Contents or Description
Version Applied	GTx executed in backward-trajectory mode (traj.79.LatLon and pm.1.010)
Model Duration	Estimation period of 168 h was considered in forward-trajectory mode Modeling duration was from 1 January to 31 December 2013
Meteorological Data	Data were obtained from the air quality monitoring network (AQMN) installed by Taiwan Environmental Protection Agency (75 sites), Taiwan Central Weather Bureau (CWB; 30 sites), and Taiwan Power Company (Taipower; 12 sites)
Emission Data	Data were obtained from the Taiwan Emission Data System (TEDS) version 9.0 (built in 2013) with five revisions: 2.5 times PM_2.5_ emission for point sources; Adding emission data from boats for line sources; Excluding diesel-fueled trucks driving on township roads for line sources; Excluding emission data from boats and suspension dust for area sources; Emission data estimates for unpaved road dust by using area and wind speed.
Altitude Meteorological Data	National Centers for Environmental Prediction (NCEP) NW & SW or CWB Banqiao & Hualien sites
Concentration from overseas	PM_2.5_ = max(PM_2.5_ at Yangmingshan AQMN minus 10 μg/m^3^, 0) and max(PM_2.5_ at Hengchun AQMN minus 5 μg/m^3^, 0); then using linear interpolation with geographic location
Major Revision	Dry deposition velocity of NOy was set as five times of NOx.

**Table 3 ijerph-19-01599-t003:** City/county based PM_2.5_ estimates in different model scenarios with coal use.

City/ County	Observed PM_2.5_ in 2013	10-Year Averaged PM_2.5_	PM_2.5_ Emitted from 5 CP Plants	PM_2.5_ Emitted from 33 CHP Plants	PM_2.5_ Emitted from the Total Coal-Fired Plants
Taipei City	24.65	26.33 (0.87)	0.60 (0.06)	0.27 (0.01)	0.87 (0.07)
Taichung City	30.93	34.60 (0.51)	1.26 (0.10)	0.64 (0.10)	1.90 (0.15)
Tainan City	35.98	38.54 (0.29)	1.63 (0.13)	0.96 (0.04)	2.59 (0.14)
Kaohsiung City	39.31	44.14 (0.34)	1.66 (0.05)	1.27 (0.06)	2.94 (0.08)
Keelung City	18.37	22.00 (0.20)	0.80 (NR)	0.29 (NR)	1.09 (NR)
Hsinchu City	30.28	30.52 (0.17)	0.86 (NR)	0.55 (NR)	1.40 (NR)
Chiayi City	**40.48**	**44.39 (0.22)**	1.68 (NR)	1.11 (NR)	2.79 (NR)
New Taipei City	25.40	26.22 (0.45)	0.86 (0.10)	0.40 (0.02)	1.26 (0.10)
Taoyuan City	25.79	27.65 (0.40)	1.01 (0.13)	0.61 (0.03)	1.62 (0.13)
Hsinchu County	25.23	27.63 (0.69)	0.80 (0.06)	0.53 (0.03)	1.32 (0.07)
Miaoli County	26.60	28.20 (0.27)	2.22 (0.74)	**1.59 (0.57)**	3.81 (0.93)
Changhua County	32.79	35.35 (0.66)	1.53 (0.10)	0.56 (0.03)	2.08 (0.11)
Nantou County	35.09	36.63 (0.57)	**4.22 (1.36)**	1.42 (0.18)	**5.64 (1.37)**
Yunlin County	32.85	35.17 (1.22)	2.04 (0.42)	1.06 (0.09)	3.10 (0.43)
Chiayi County	32.93	37.15 (0.57)	1.58 (0.05)	1.00 (0.12)	2.58 (0.13)
Pingtung County	32.87	33.33 (3.39)	1.15 (0.29)	0.72 (0.07)	1.87 (0.30)
Ilan County	19.55	19.56 (0.29)	0.42 (0.10)	0.19 (0.09)	0.61 (0.13)
Hualien County	18.40	20.14 (0.19)	0.74 (NR)	0.03 (NR)	0.77 (NR)
Taitung County	13.09	13.66 (0.14)	0.19 (NR)	0.13 (NR)	0.32 (NR)
Taiwan	28.45	30.59 (4.05)	1.33 (0.89)	0.70 (0.45)	2.03 (1.29)

Standard deviation in parentheses; Unit in μg/m^3^; NR: Not Report. The maximum value is shown in bold.

**Table 4 ijerph-19-01599-t004:** Statistics of numbers of county, AQMN, and maximum affected distance from each coal-fired plant (CP and CHP) with greater than 0.1 μg/m^3^ PM_2.5_ increment.

Plant No.	No. of County	No. of AQMN	Max Affected Distance ^1^
CP#01	8	30	174 km
CP#02	16	66	193 km
CP#03	6	18	155 km
CP#04	8	35	272 km
CP#05	4	12	183 km
CHP#01	0	0	0 km
CHP#02	0	0	0 km
CHP#03	0	0	0 km
CHP#04	0	0	0 km
CHP#05	0	0	0 km
CHP#06	0	0	0 km
CHP#07	0	3	18 km
CHP#08	0	1	8 km
CHP#09	0	0	0 km
CHP#10	5	11	96 km
CHP#11	0	1	4 km
CHP#12	0	1	11 km
CHP#13	0	6	23 km
CHP#14	0	0	0 km
CHP#15	0	1	3 km
CHP#16	0	3	10 km
CHP#17	0	1	1 km
CHP#18	0	0	0 km
CHP#19	2	13	31 km
CHP#20	0	0	0 km
CHP#21	0	0	0 km
CHP#22	0	0	0 km
CHP#23	6	17	151 km
CHP#24	0	0	0 km
CHP#25	0	0	0 km
CHP#26	1	2	11 km
CHP#27	6	18	157 km
CHP#28	1	2	8 km
CHP#29	0	2	10 km
CHP#30	4	11	107 km
CHP#31	1	2	8 km
CHP#32	1	1	2 km
CHP#33	1	1	7 km

^1^ Max Affected Distance: Maximum affected distance from CP/CHP source to AQMN. Abbreviation: AQMN: Air quality monitoring network; CP: Coal-fired power plant; CHP: Combined heat and power plant.

## Data Availability

Not applicable.

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
