# Peer review of "County-Wide Mortality Assessments Attributable to PM2.5 Emissions from Coal Consumption in Taiwan"

_ijerph, 2022, doi:10.3390/ijerph19031599_

Round 1

Reviewer 1 Report

General comments

*The authors should review more papers to support that coal power plants are one of the main contributor of PM2.5 in Taiwan.

Specific comments

Line-55-56: The coal-fired process may not only emit particulate matter (PM), sulfur oxides (SOx), nitrogen oxides (NOx), carbon dioxide (CO2), and volatile organic compounds but also including some POPs such as PAHs, dioxins and furans. Please carefully review the following articles and cite them.

Pongpiachan, S., Wiriwutikorn, T., Rungruang, C., Yodden, K., Duangdee, N., Sbrilli, A., ... & Centeno, C. (2016). Impacts of micro-emulsion system on polychlorinated dibenzo-p-dioxins (PCDDs) and polychlorinated dibenzofurans (PCDFs) reduction from industrial boilers. Fuel, 172, 58-64.

Pongpiachan, S., Wiriwutikorn, T., Sbrilli, A., Gobbi, M., Hashmi, M. Z., & Centeno, C. (2019). Influence of Fuel Type on Emission Profiles of Polychlorinated Dibenzo-p-Dioxins and Polychlorinated Dibenzofurans from Industrial Boilers. Polycyclic Aromatic Compounds.

Line-60-61: What type of industrial boilers used in coal power plants in Taiwan (e.g. supercritical boiler or subcritical boiler)? Please be specific.

Line-64-65: A previous study highlight the importance of CO as one of the most influential gaseous species responsible for the hospital admission. Please carefully review some previous articles related to this issue.

Line-90-91: Instead of using the Weather Research and Forecasting Model coupled with Chemistry (WRF-Chem), the authors selected GTx model for forecasting PM2.5. Any particular reasons? Please explain.

Line-115-116: More details related to the computation of the fine fractions of PM10 (i.e. 0.85 and 0.58) are essentially required.

Line-181: There are several points that need more explanation in Fig. 2. Firstly, the authors should explain the reasons why the model overestimated PM2.5 in the northern and north-eastern region of Taiwan? Secondly, the model underestimated PM2.5 levels in the western part of Taiwan. Why?

Line-202: The authors should provide more explanations associated with the information displayed in Fig.3. It appears that the distribution of PM2.5 over the past ten years show similar patterns with those of GTx modeled PM2.5 estimated from all coal fired power plants. On the contrary, the population distribution did not match with the distribution of PM2.5, which means anthropogenic activities (e.g. traffic emission and domestic heating) are not the main source of PM2.5. These findings are not in good agreement with previous studies related to source apportionment in Taiwan. For instance, Tsai et al (2020) indicated that road dust is one of the main contributor of PM2.5 (https://www.sciencedirect.com/science/article/abs/pii/S1309104220300234).  Lu et al (2016) considered that traffic emission is the most dominant primary source in even a coastal city (https://aaqr.org/articles/aaqr-16-01-oa-0008). How the authors will explain this discrepancy?

Line-234: The authors consider the distance factor from each coal fired plant to air quality monitoring stations as one of the most influential parameters of governing PM2.5 level. What about the other factors such as Planetary Boundary Layer (PBL), wind speed, wind direction, precipitation and other meteorological factors?

Reviewer 2 Report

Chio et al. first estimated the county-level PM2.5 concentration in Taiwan via a forward model and evaluated it against observations. They further estimated the number of counties and air quality stations that are affected by the coal-fired or combined heat and power plants, as well as the annual adult and premature deaths associated with those plants. Such a subject can be important to both the scientific community and society.

However, the current presentation requires a lot more effort, since many technical details are missing and some texts/sentences are very awkward, which greatly impair the understanding and evaluation of their method and the assessment of their quantitative results.

I had a hard time understanding how the authors related mortality to PM2.5 concentration and air quality stations (Sect 2.2 and 2.3 or equation 5), which is the key of this paper. For example, what are “RR estimates” (L148) and how does “RR” relate to the PM2.5 concentration (C(i))? Why is the "linear assumption between PM2.5 exposure and disease outcomes" (L146) a reasonable assumption? How was the “maximum (affected) distance” (e.g., L213, L216) being calculated? Any consideration of the wind direction between the power plants and the air quality stations?

Even though prior studies might cover the material, the authors should be responsible for providing sufficient technical details and explanations so that any readers can understand.

In addition, the current presentation contains many grammatical mistakes (e.g., “model PM2.5 findings”(L178) -> modeled) and awkward expressions/sentences (e.g., L207 – “there were between...”; “concerns” on L138). I would recommend making the text clearer and going through grammatical checks if possible. 

Reviewer 3 Report

The paper “County-wide mortality assessments attributable to PM2.5 emissions from coal consumptions in Taiwan” [ijerph-1533401] has been submitted for publication in the “International Journal of Environmental Research and Public Health”. The paper is very well conceived, and also well written. Only a couple of points need to be cleared before being eventually ready for publication:

1. Transportation is generally considered to be one of the main PM2.5 polluting sectors. Yet, contributions deriving from such sector have been totally ignored in this interesting paper. I suggest the authors to add a brief paragraph in their introduction, quoting the following high-impact papers: Bruzzone, F., et al. (2021). The combination of e-bike-sharing and demand-responsive transport systems in rural areas: A case study of Velenje. Research in Transportation Business & Management 40, 100570; Jandacka, D., et al. (2017). The contribution of road traffic to particulate matter and metals in air pollution in the vicinity of an urban road. Transportation Research Part D: Transport and Environment 50, pp. 397-408; Nocera, S., et al. (2018). Options for Reducing External Costs from Freight Transport along the Brenner Corridor. European Transport Research Review 10(2), 53;

  1. In the conclusions, the authors should expand on the generalization potential of their research. What would happen if their method should be applied outside of Taiwan?

All in all, this is an above-average paper that can be considered for publication from this journal, when both these points are fixed

Round 2

Reviewer 2 Report

The authors have revised the manuscript accordingly, especially providing more technical details that facilitate the readers. 

Reviewer 3 Report

Points all cleared. Publish